# A Changepoint Detection-Based General Methodology for Robust Signal Processing: An Application to Understand Preeclampsia’s Mechanisms

**DOI:** 10.3390/bioengineering12060675

**Published:** 2025-06-19

**Authors:** Patricio Cumsille, Felipe Troncoso, Hermes Sandoval, Jesenia Acurio, Carlos Escudero

**Affiliations:** 1Vascular Physiology Laboratory, Department of Basic Sciences, Universidad del Bío-Bío, Chillán 3780000, Chile; fetronc@gmail.com (F.T.); hermes_2990@hotmail.com (H.S.); jeseniacurio2@gmail.com (J.A.); 2Centre for Biotechnology and Bioengineering (CeBiB), University of Chile, Santiago 8370456, Chile; 3Group of Research and Innovation in Vascular Health (GRIVAS Health), Chillán 3780000, Chile

**Keywords:** brain blood perfusion, preeclampsia, experimental model, offspring from preeclampsia, changepoint detection, signal processing, computer calculation

## Abstract

Motivated by illuminating the underlying mechanisms of preeclampsia, we develop a changepoint detection-based general and versatile methodology that can be applied to any experimental model, effectively addressing the challenges of high uncertainty produced by experimental interventions, intrinsic high variability, and rapidly and abruptly varying time dynamics in perfusion signals. This methodology provides a systematic and reliable approach for robust perfusion signal analysis. The main innovation of our methodology is a highly efficient automatic data processing system consisting of modular programming components. These components include a signal processing tool for optimal segmentation of perfusion signals by isolating their “genuine” vascular response to experimental interventions, and a novel and suitable normalization to evaluate this response concerning an experimental reference state, typically basal or pre-intervention. In this way, we can identify anomalies in an experimental group compared to a control group by disaggregating noise during the transitions just after experimental interventions. We have successfully applied our general methodology to perfusion signals measured from a preeclampsia-like syndrome model developed by our research group. Our findings revealed impaired brain perfusion in offspring from preeclampsia, particularly dysfunctional brain perfusion signals with inadequate perfusion signal vasoreactivity to thermal physical stimuli. This general methodology represents a significant step towards a systematic, accurate, and reliable approach to robust perfusion signals analysis across various experimental settings with diverse intervention protocols.

## 1. Introduction

In recent years, advances in imaging technologies have provided real-time responses and continuous brain microcirculation perfusion monitoring during more extended periods. In particular, the laser speckle contrast imaging (LSCI) technique has optimized brain blood flow measurement due to high spatial and temporal resolutions. However, analysis of brain perfusion signals in the in vivo context is quite complex and even harder when analyzing experimental data from animal models for several reasons. The main reason is that brain perfusion measurements have some implicit artifacts occurring during experimentation, such as (1) high intrinsic individual variability of the microcirculation, (2) differences in time frame analysis, and (3) considerable signals’ noise generated by instrumental interventions (mostly manual). Thus, when analyzing brain perfusion, one risks partly disregarding the feasible biological response by not dealing with data uncertainty induced by the significant noise of the signals. In addition, the LSCI technique generates significant amounts of data that require sophisticated analysis techniques, making researchers invest more effort in processing. Therefore, robust methods are crucial to analyzing brain perfusion signals properly.

Regarding brain perfusion measurements, we have progressed in animals from two different conditions using mice exposed to preeclampsia [1]. Preeclampsia (PE) is a human pregnancy hypertension syndrome characterized by reduced placental perfusion and maternal endothelial dysfunction present after 20 weeks of gestation [2,3]. This maternal complication is associated with harmful consequences in the offspring, including a high risk of developing neurological, psychological, or behavioral disorders [4,5,6,7]. We would like to emphasize that research into brain complications and long-term neurodevelopmental outcomes in offspring exposed to PE is still an evolving field. Only a few systematic reviews and meta-analyses have addressed this topic [6,8,9,10]. In [6], it was reported that while the pooled unadjusted estimates did not support a statistically significant association between maternal PE and autism spectrum disorder (ASD) or attention-deficit/hyperactivity disorder (ADHD), the adjusted odds ratios suggested an increased risk, approximately 50% for ASD and 28% for ADHD. These findings contrast with a more recent meta-analysis showing significant associations, indicating a 27% increased risk of ASD and a 29% increased risk of ADHD in offspring exposed in utero to PE [10]. Summarizing, recent meta-analyses report a 27–50% increased risk of ASD and ADHD in children exposed to PE in utero [6,10]. The differences in these studies highlight the evidence’s growth and still-debated nature linking PE to long-term neurodevelopmental disorders. Given the heterogeneity in study designs, diagnostic criteria, and confounder adjustments across studies, further high-quality longitudinal research is required to delineate these associations more precisely.

Preclinical studies in rodents, including pups born from PE-like syndrome (PELS) [11,12], or even clinical studies involving children born to women with PE, have reported impaired brain structural alterations in their children compared with children born to mothers without hypertension in pregnancy [13,14,15,16]. Notably, studying offspring of PELS generated by three different models, we have shown a reduction in the brain angiogenesis process [1] or in the brain microvascular perfusion in a model of PELS generated by administration of the nitric oxide inhibitor, N(ω)-nitro-L-arginine methyl ester (L-NAME) [17]. In addition, impaired brain perfusion seems to be more preferentially present in the female offspring [17,18].

Regarding robust analysis methods, the potential for innovation in optimal changepoint detection-based signal processing applied to brain microvascular blood perfusion analysis is immense. A search for citations of the seminal paper by [19]—which introduced the *Pruned Exact Linear Time (PELT)* method, which is highly efficient for optimal signal segmentation, refined by “signal processing” and “brain” within all fields—in the Web of Science (WOS) yielded only eight results. To ensure reproducibility, the search was as follows. First, we looked for “Optimal Detection of Changepoints With a Linear Computational Cost” in Journal Citation Reports provided by the WOS. Second, we clicked on this title to view its citations. Third, we refined the citations twice by using the menu “search within all fields” by writing “Signal Processing” followed by “Brain”, regardless of years and keywords. We do not aim to discuss the eight citations here. Still, we limit ourselves to highlighting that this scarcity of works devoted to signal processing for brain perfusion analysis underscores the untapped potential in this area, which should inspire and motivate researchers to make groundbreaking advancements and open new perspectives.

In our previous work [18], which forms part of the eight citations of [19] devoted to signal processing for brain perfusion analysis, we developed a signal processing tool that consisted of optimal segmentation and computation of least-squares piecewise linear approximations (PLAs) for perfusion signals. However, our focus was not on precisely segmenting perfusion signals by disaggregating noise produced by experimental interventions. In this regard, in the present work, we define the times of interest (TOIs) as the times of measurements when the perfusion signal is free of noise produced by experimental interventions. In [18], we did not necessarily disaggregate noise between the TOIs or during the transitions just after experimental interventions. We compared brain blood perfusion signals, measured by an LSCI technique, of a control (or wild type, WT) group and an experimental one, the latter being a reduced uterine perfusion pressure (RUPP) model for PELS. Despite our advances in finding optimal segmentation and making precise calculations of PLAs, we acknowledge some limitations, such as artifacts generated by the administration of physical stimuli. These artifacts induced much noise in perfusion signals, prompting us to improve our methodology to isolate the ”genuine” vascular response to stimuli. It is crucial to systematically address these technical and experimental limitations in brain perfusion analysis, which is critical for reducing imprecision in elucidating fundamental biological questions, thus encouraging advances in the field.

Our overarching goal is to systematize, improve, and generalize our signal processing tool [18] to unravel the “genuine” vascular response, defined as the biological response of the brain’s vasculature to a given experimental intervention, free of the noise produced by the latter. So, the main strength of our *general methodology* is that it addresses technical and experimental limitations by disaggregating noise between the TOIs. In addition, it is versatile since we can apply it to analyze microvascular blood perfusion signals of any nature without altering the main components of the methodology. These properties make it a valuable tool for a solid quantitative analysis of brain perfusion signals, thereby illuminating, in particular, the complex mechanisms of preeclampsia.

## 2. Materials and Methods

Our general methodology’s main innovation is a highly efficient automatic data processing system consisting of modular programming components. These components include a signal processing tool for optimal segmentation in TOIs and a novel normalization. These components can isolate the genuine vascular response of perfusion signals to experimental interventions from an experimental reference state, usually basal or pre-intervention. Thus, our general methodology can accurately and reliably compare brain perfusion signals between an experimental and a control group.

The optimal segmentation in TOIs and the novel normalization constitute the main contribution of our general methodology and are described in Section 2.3. They identify the most relevant perfusion signal segments from the experimental and statistical viewpoints, excluding the noise produced by experimental interventions, and removing any systematic biases for accurate and reliable comparison.

The general methodology starts with a data preprocessing step that consists of data collection and formatting following a given experimental model, whose datasets correspond to perfusion signals measured in TOIs determined by the experimenter (yet not optimal), which consist of a basal or pre-intervention state and states after experimental interventions.

In the remainder of this section, we conceptually present the general methodology, the implementation details of which are in Appendix B.

### 2.1. Data Preprocessing

Data preprocessing is the only step in the general methodology that requires manual intervention from the experimenter and differs according to each experimental model. The experimenter’s role in this step is to conduct an experimental model that involves making perfusion signal measurements in TOIs (a basal or pre-intervention and after experimental interventions). However, from the overall algorithm’s perspective, we strictly limit the experimenter’s role to data collection and transcription, ensuring that we accurately and consistently record the data.

The output of data preprocessing consists of an Excel file prepared by the experimenter. This file contains the raw data used in the subsequent methodology steps. As explained in Section B.1, the file is formatted in four columns for every experimental subject.

Once the Excel file is formatted, the first two steps of the general methodology are as follows:Step 1: Read the data preprocessed in the Excel file and convert it into matrices.Step 2: Save the matrices into two data files, one for each group, control and experimental.

In this work, we analyze datasets from a PELS model developed by our research group to illuminate the underlying mechanisms of PE. We refer to Appendix B for details on the experimental model and motivation for the general methodology development. We do not describe the experimental model in the main body of this article since it has already been implemented and published in [1]. In contrast, in this article, we focus on the general methodology and its application to conduct a robust brain perfusion signal analysis to establish an accurate and reliable comparison between two subject groups.

### 2.2. Automatic Data Processing

It consists of the four stages that follow the first two described in the previous section:Step 3: Format the matrices into as many columns as experimental TOIs.Step 4: Compute the optimal segmentation in TOIs and normalize perfusion signals; Section 2.3 and Section B.3.Step 5: Calculate the matrices that one wants to compare.Step 6: Perform the comparative statistical analysis; Section B.5.

### 2.3. Calculation of the Optimal TOIs and Optimal Transitions

Considering experimental TOIs (those determined by the experimenter, not yet optimal), we compute optimal data segmentation in TOIs, which produces two matrices: one that stores the times associated with every TOI, and the second one that stores the transition times between the TOIs. We calculate both matrices for every dataset associated with each experimental subject.

To calculate optimal segmentation in TOIs and transition regions between the TOIs, we considered the “markers” column (the third one of the Excel file), where the experimenter indicated the TOIs each measurement pertains to (basal and after interventions). Of course, the experimenter defined these markers in an ad hoc way, following the experimental protocol. This definition is subject to errors due to implicit artifacts that usually occur during experimentation: (1) high individual variability of the microcirculation, (2) biologically relevant differences in time frame analysis, and (3) artifacts or considerable signal noise (mostly manual) generated by instrumental intervention. Therefore, data segmentation in optimal TOIs is a statistical tool for accurately and reliably comparing each perfusion signal’s response to experimental interventions. In practice, however, the perfusion signal for a given individual cannot be segmented. Perfusion signal segmentation is a mathematical artifact to achieve the work’s comparison goal.

To overcome the issues (1)–(3), we designed the original Algorithm 1. To implement it, we first apply the pure PELT method, i.e., signals’ segmentation without calculating yet optimal TOIs and without disaggregating noise between the TOIs or during the transitions just after experimental interventions. The main output argument of the pure PELT method is a vector containing the positions of changepoints, where the PLA change most significantly for every perfusion signal, denoted by $\tau \in R^{m+2}$; Section B.3. Then, we split τ into $n_{T}$ vectors denoted by $\tau^{1},\ldots ,\tau^{n_{T}}$, where $n_{T}$ stands for the total number of TOIs. Each vector $\tau^{j}$ contains the positions of changepoints for every TOI $j=1,\ldots ,n_{T}$, which we construct by intersecting the “markers” column associated with each dataset collected in the Excel file, with vector B(τ,1) that contains the times of occurrence of changepoints. Here *B* is a matrix of size $n_{i}\times 2$, $n_{i}$ being the number of measurements for experimental subject *i*, whose columns 1 and 2 store the time points and perfusion values, which we read from the second and fourth columns associated with each studied experimental subject in the Excel file; Section B.1.

Considering as input arguments vectors $\tau ,\;\tau^{1},\;\ldots \;,\;\tau^{n_{T}}$, Algorithm 1 reads as follows.
**Algorithm 1:** Optimal TOIs and transition times.   1:**Inputs:** $\tau ,\;\tau^{1},\;\ldots \;,\;\tau^{n_{T}}$   2:**Outputs:** OTOI,OTT   3:$k\leftarrow \mathrm{length}\left[\tau^{2},\;\ldots ,\tau^{n_{T}}\right]=\mathrm{length}\left[\tau \setminus \tau^{1}\right]$   4:**for** $j=1,\ldots ,n_{T}$**do**   5:    $IT_{j}\leftarrow \min\left(B(\tau^{j},1)\right)$   6:    **if** j=1 **then**   7:        $OTOI_{•,1}\leftarrow {\left[B\left(1,1\right),\;\ldots \;,IT_{2}-k-1\right]}^{T}$   8:        $OTT_{•,1}\leftarrow {\left[IT_{2}-k,\;\ldots \;,IT_{2}+k\right]}^{T}$   9:    **else if** $j=n_{T}$ **then** 10:        $OTOI_{•,n_{T}}\leftarrow {\left[IT_{n_{T}}+k+1,\;\ldots \;,B\left(n_{i},1\right)\right]}^{T}$ 11:    **else** 12:        $OTOI_{•,j}\leftarrow {\left[IT_{j}+k+1,\;\ldots \;,IT_{j+1}-k-1\right]}^{T}$ 13:        $OTT_{•,j}\leftarrow {\left[IT_{j+1}-k,\;\ldots \;,IT_{j+1}+k\right]}^{T}$ 14:    **end if** 15:**end for**

The output arguments of Algorithm 1 are two matrices, OTOI and OTT, where OTOI contains the “optimal TOIs” considering noise disaggregation between the TOIs and OTT the “optimal transition times” between the TOIs.

The key of Algorithm 1 is to compute vector $IT\in R^{n_{T}}$ that contains the experimental “intervention times” defined as the minimum of the times of occurrence of changepoints for every TOI *j*, as computed by the pure PELT method. Our Algorithm 1 computes the matrix OTOI separately from the matrix OTT that stores the optimal transition times between the optimal TOIs. By doing so, we effectively disaggregate considerable signal noise from instrumental interventions. Consequently, we can compare the genuine response of perfusion signals after their stabilization, focusing solely on their intrinsic variability, not noise produced by interventions during transitions before stabilization.

We remark that the *k* value assigned in line 3 of Algorithm 1, which we subtract and add from the experimental intervention times (vector IT) in lines 8 and 13, ensures an accurate and reliable optimal segmentation of perfusion signals. In addition, given that *k* is defined as the length of the changepoints in the times of interest (TOIs) after experimental interventions or basal TOI, it only depends on when the experimenter applied experimental interventions and not the dataset itself. This makes our general methodology versatile since it applies to any experimental setting.

### 2.4. Data Normalization

To elucidate whether experimental interventions induce significant changes in perfusion, we normalize perfusion signals concerning a reference state for every experimental subject. To do so, we describe perfusion signals as discrete-time functions representing experimental measurements. Precisely, (1)$$\begin{array}{c} {\left(y_{t}^{i,j}\right)}_{t^{i,j-1}+1\leq t\leq t^{i,j}} \end{array}$$(2)$$\begin{array}{c} {\left(y_{t}^{i}\right)}_{1\leq t\leq n_{i}}={\left({\left(y_{t}^{i,j}\right)}_{t^{i,j-1}+1\leq t\leq t^{i,j}}\right)}_{j=1,\ldots ,n_{T}} \end{array}$$

Equation (Equation 1) represents the perfusion signal in the optimal TOI or segment *j*, $t^{i,j}$ is the end time for every TOI or segment *j*, $t^{i,0}=0$, (and, respectively, $t^{i,n_{T}}=n_{i}$), which corresponds to the first (and, respectively, final) measurement, $n_{i}$ being the number of measurements for experimental subject *i*. Equation (2) describes the entire perfusion signal for experimental subject *i*. With these notations, we can introduce a novel *z*-*normalization* for every perfusion signal or its PLA associated with each experimental subject *i*, and every TOI *j*. This normalization is defined by(3)$$z_{t}^{i,j}=\frac{y_{t}^{i,j}-\bar{y^{i,\mathrm{ref}}}}{s^{i,\mathrm{ref}}}\;\mathrm{and}\;{\hat{z}}_{t}^{i,j}=\frac{{\hat{y}}_{t}^{i,j}-\bar{{\hat{y}}^{i,\mathrm{ref}}}}{{\hat{s}}^{i,\mathrm{ref}}}\;.$$

In Equation (Equation 3), we normalize every perfusion signal value $y_{t}^{i}$, $t=1,\ldots ,n_{i}$ in Equation (2), for every TOI $j=1,\ldots ,n_{T}$ and every experimental subject i=1,…,N concerning a reference state ‘ref’, usually the basal state j=1. The quantities $\bar{y^{i,\mathrm{ref}}}$ and $s^{i,\mathrm{ref}}$ represent the mean and standard deviation of the perfusion signal for the experimental subject *i* in the reference state. The same explanation is valid replacing $y_{t}^{i}$ for its PLA ${\hat{y}}_{t}^{i}$, $t=1,\ldots ,n_{i}$, defined in Equation (Equation 10). This way, we can statistically compare perfusion signals after experimental interventions ($j=2,\ldots ,n_{T}$) suitably concerning a basal/reference state (j=1) among experimental subjects (i=1,…,N), as we show below.

We highlight that any statistical test to compare two or more datasets normalizes every data point concerning its sample mean but not concerning a reference state, as made in the present study. From Equation (Equation 3), a simple calculation shows that for every $k,\ell =2,\ldots ,n_{T}$ with k≠ℓ, one has(4)$$\left|z_{t}^{i,k}-z_{t}^{i,\ell }\right|=\frac{\left|y_{t}^{i,k}-y_{t}^{i,\ell }\right|}{s^{i,\mathrm{ref}}}\;.$$

**Remark 1.** 
*Despite the number of measurements in the TOIs k,ℓ being generally different, we format the corresponding data segments in a suitable matrix for every experimental subject. This way, we can handle the corresponding data segments for comparison; see Section B.2 for details.*


Consequently, if k,ℓ>1 are two distinct TOIs after interventions, and $\left|z_{t}^{i,k}-z_{t}^{i,\ell }\right|$ is greater than a certain threshold for *i* fixed, according to Equation (Equation 4), this implies that perfusion signal in TOIs or segments j=k and j=ℓ defined in Equation (Equation 1), have mean and variance significantly different of those given by the reference TOI j=1. In other words, perfusion signal segments in experimental states k,ℓ>1 would have significant differences concerning the corresponding basal perfusion signal for experimental subject *i*. Therefore, with normalization (Equation 3), we can establish or deny significantly different responses of perfusion signals to any experimental intervention concerning the basal/reference state. This normalization process is crucial as it could help elucidate anomalies in the perfusion signals’ response to experimental interventions of an experimental group compared to a control one for any experimental model.

From a physiological perspective, driven by the goal of uncovering the underlying mechanisms of PE, our approach can significantly enhance understanding of brain blood perfusion in offspring from normal pregnancies compared with the PELS. This understanding could help tackle the high risk of long-lasting cognitive consequences and stroke in the offspring of PE.

## 3. Results

In this section, we illustrate the general methodology application to the experimental model data described in Appendix B, addressing the significant research questions that have shaped our approach. In that experimental model, the total number of experimental subjects is N=33 and the total number of TOIs is $n_{T}=3$. The TOI j=1 corresponds to the basal state or pre-intervention, the TOI j=2 to the cold state or after the cool stimulus (first intervention), and the TOI j=3 to the warm state or after the warm stimulus (second intervention). The experimental subject i=1,…,16 belongs to the control or WT group, whereas i=17,…,N to the experimental or L-NAME group. To clarify notations for the experimental model we analyze here, we follow the general notation (2) to provide some examples. For instance, (5)$$\begin{array}{ccc} {\left(y_{t}^{16,1}\right)}_{t=1,\;\ldots \;,t^{16,1}} & & {\left(y_{t}^{23,2}\right)}_{t=t^{23,1}+1,\;\ldots \;,t^{23,2}} \end{array}$$(6)$$\begin{array}{ccc} {\left(z_{t}^{16,1}\right)}_{t=1,\;\ldots \;,t^{16,1}} & & {\left(z_{t}^{23,2}\right)}_{t=t^{23,1}+1,\;\ldots \;,t^{23,2}} \end{array}$$

Equation (Equation 5) designates the perfusion signal segment measured for individual i=16 (WT male) in the basal state or TOI j=1, and the perfusion signal segment for individual i=23 (L-NAME female) in the cold state or TOI j=2, while Equation (6) stands for the corresponding normalized perfusion signal segments according to normalization (Equation 3), and so on.

Figure 1 and Figure 2 depict the application of the general methodology to datasets $y_{t}^{i}$, $1\leq t\leq n_{i}$, defined in Equation (2) for mouse i=9 (WT male), and mouse i=17 (L-NAME female), respectively. This application provides a visually compelling comparison, as we plot perfusion signals and their respective PLAs in Figure 1 and Figure 2 on the same scales for the two axes. Precisely, we plot(7)$${\left(z_{t}^{i}\right)}_{1\leq t\leq n_{i}}\;\mathrm{and}\;{\left({\hat{z}}_{t}^{i}\right)}_{1\leq t\leq n_{i}},$$
which are the normalization computed according to Equation (Equation 3) for perfusion signal $y_{t}^{i}$, $1\leq t\leq n_{i}$ and its corresponding PLA ${\hat{y}}_{t}^{i}$, $1\leq t\leq n_{i}$, defined by Equation (Equation 10) and described in Section B.3. We plot datasets given in Equation (Equation 7) segmented in the optimal TOIs, as denoted by Equation (Equation 1). In addition, we plot the optimal transition times between the optimal TOIs, shown in Figure 1 and Figure 2 as segmented vertical lines.

Figure 1 and Figure 2 demonstrate the robustness of our general methodology. The accurate numerical segmentation calculation into the three optimal TOIs, optimal transition times between the TOIs, and the normalized perfusion signal’s PLA corroborate its strength.

Physiologically, we observe basal perfusion with slightly more intrinsic variability for the L-NAME female offspring (in some regions). Importantly, its perfusion signal does not respond to thermal stimuli (Figure 2). On the contrary, the WT male offspring underwent a notorious response to the thermal stimuli, showing an overall drop in perfusion values after the cool stimulus and high intrinsic variability and an overall rise (or recovery concerning its basal state) in perfusion values after the warm stimulus (Figure 1).

The visual comparison in Figure 1 and Figure 2 reveals relevant differences in the perfusion signals of both individuals. In this regard, previous reports have described impaired capacity of brain blood vessels to respond to systemic blood pressure (i.e., autoregulation). Therefore, we would like to know whether these alterations in the middle/large brain blood vessels could also affect brain microcirculation [20,21]. We will quantitatively study this kind of alteration in perfusion signals for the two mouse groups and sexes, as shown in Figure 1 and Figure 2. Indeed, the qualitative result shown in Figure 1 and Figure 2 is a general finding that characterizes perfusion signals for the two mouse groups and sexes. To show that, we apply our general methodology, whose essence is its ability to compare the genuine response of perfusion signals to experimental interventions reliably and accurately. This comparison is reliable since we exclude the optimal transition times between the optimal TOIs (Figure 1 and Figure 2). By doing so, we effectively eliminate considerable signal noise generated by experimental interventions, such as thermal physical stimuli. Consequently, we can compare the genuine response of perfusion signals after their stabilization, focusing solely on their intrinsic variability, not noise produced during transitions. In addition, the comparison is accurate since we quantify the response of perfusion signals to experimental interventions concerning their basal/reference or pre-intervention state, by applying the novel normalization defined in Equation (Equation 3).

The rest of this section is organized as follows. In Section 3.1 below, we compare perfusion signals in the basal state, while in Section 3.2 and Section 3.3, we conduct a statistical study to support the result shown in Figure 1 and Figure 2 as a general finding.

### 3.1. Comparative Statistical Analysis in Basal State

This section compares perfusion signals for experimental subjects and the four interest groups (WT females, WT males, L-NAME females, and L-NAME males) in the basal/reference state. We also compare normalized cold-to-basal and warm-to-cold differences for the four interest groups.

To compare and better visualize, we only once depict the non-normalized basal perfusion signals for experimental subjects in Figure 3. From it, one seems to observe individual variability.

However, the K-W test revealed a *p*-value of 0.5903 for individuals in Figure 3, indicating no significant differences among non-normalized basal perfusion signals. As one could expect, the normalized basal perfusion, depicted in Figure 4 (the first four bars), does not show significant differences either (K-W test, p=0.1418).

Physiologically, our results suggest that the basal brain perfusion signals do not significantly differ between individuals and the four interest groups.

### 3.2. Comparative Statistical Analysis for Cold-Induced Response

As explained in Appendix B, in the experimental model developed by our research group, the experimental interventions correspond to thermal physical stimuli (cool and warm) applied to the offspring of mice from two groups: control (WT) and experimental (L-NAME).

Next, we depict the cold-to-basal differences for the normalized perfusion signals, for raw data in Figure 5 and for PLAs in Figure 6. Precisely, both figures show box and whisker plots, one for each experimental subject i=1,…,N, for the differences $z_{t}^{i,2}-z_{t}^{i,1}$ in Figure 5 and for the differences ${\hat{z}}_{t}^{i,2}-{\hat{z}}_{t}^{i,1}$ in Figure 6, respectively. Normalization ${\left(z_{t}^{i}\right)}_{1\leq t\leq n_{i}}$ and ${\left({\hat{z}}_{t}^{i}\right)}_{1\leq t\leq n_{i}}$ are defined by Equation (Equation 3).

Notably, the normalized values of female and male WT mice showed more dispersion (wider range of values) than female and male L-NAME mice. When we plotted cold-induced perfusion response per group (from the fifth to the eighth bar in Figure 6), WT’s offspring responded significantly differently than L-NAME’s offspring to the cool stimulus.

Our results underscore the importance of our research, indicating that 50% of WT’s offspring, as represented by the median, show an overall drop in the brain perfusion signals after the cool stimulus. This finding is particularly relevant as it is consistent with the case represented by Figure 5 for offspring 9 (WT male, internal code WT14D4). On the other hand, offspring from PE did not respond to the cool stimulus.

Our research confirmed the individual differences in response to the cool stimulus. The K-W test applied to normalized data for the separate individuals depicted in Figure 5 (K-W test, $p<1\times 10^{-10}$), and the pairwise multi-comparison test applied to the individual cold-to-basal state differences produced the results reported in Table 1, validating the individual differences in response to the cool stimulus.

Table 1 showcases the top 10 individuals with remarkable variations in their cold-induced response. These individuals were selected based on their unique responses compared to at least 27 other subjects we analyzed. Notably, the top six individuals are from the WT group, with five of the six being male.

These results highlight the significant individual differences in response to the cool stimulus, a crucial aspect that piques our interest in this research. Notably, the perfusion signals of mice 9 and 16 (both male WT) stand out as significantly different from all other analyzed mice. We rigorously tested these individual differences to determine whether they were due to specific individuals or the entire interest group. Table 1 reveals that several individuals have significant inter- and intra-group differences, indicating that these differences are not specific to individuals but to the entire interest groups.

Now, we question whether the previous results are due to the high individual variability of raw perfusion signals. Since the PLAs have less variability than raw data, these numerical approximations could indicate a more realistic difference among the individuals studied.

Our preference for the entire PLAs over their slopes or averages, as previously published [18], is rooted in the methodology’s ability to provide optimal segmentation. This segmentation allows for accurate separation in experimental states (i.e., in TOIs) and useful approximations for every perfusion signal, which is valuable for deeper analysis. Even without approximating perfusion signals, our methodology is necessary for accurately separating the TOIs and, thus, precisely quantifying the perfusion vascular response of offspring to both stimuli.

Figure 6 presents box and whisker plots for the normalized PLAs. Despite less variability, Figure 6 shows that PLAs behave qualitatively similarly to the raw cold-to-basal perfusion differences (Figure 5). This analysis confirms that WT offspring undergo an overall diminution, whereas L-NAME offspring show a slight overall increase in brain perfusion to the cool stimulus (Figure 6). We further confirmed these differences by applying the K-W test for the PLAs for individuals depicted in Figure 6. The K-W test for separate individuals yielded $p<1\times 10^{-10}$, indicating a significant difference. It provides strong evidence for the observed differences and reassures the reader of the accuracy of our results.

As expected, the cool stimulus reduced brain perfusion in female and male WT pups. However, offspring from L-NAME-treated dams exhibited no response to the cool stimulus. We did not study the underlying mechanisms responsible for this alteration here. However, previous research showed that offspring from PE have impaired formation [1] and function [21] of brain blood vessels. The present work also indicates alterations also in the microcirculation. This lack of response to cold-induced vasoconstriction in offspring of PE indicates functional alterations that may prompt future cognitive or cerebrovascular diseases. Results also indicate that the male response in each group was exacerbated compared with the female response. Thus, males in the WT group exhibited an increased drop, whereas in the L-NAME group, males exhibited significantly higher brain perfusion than their female siblings. Currently, increasing evidence indicates sex-specific alterations associated with pregnancy complications, including PE [22]. This evidence indicates that males are more susceptible to alterations in the cardio-cerebrovascular or metabolic systems. Our results align with this evidence, indicating that male offspring of PE have brain blood vessels that anomalously respond to cold-induced vasoconstriction. The causes and consequences of this defect require further investigation.

### 3.3. Comparative Statistical Analysis for Warm-Induced Response

As in our previous analysis, we meticulously and rigorously compared the warm-induced vascular response of normalized perfusion signals depicted in Figure 7 and Figure 8, which represent perfusion signals and their PLAs, respectively, instilling confidence in the thoroughness and reliability of our general methodology.

Our general methodology revealed distinct responses between the WT and L-NAME offspring. When we grouped the data into the four interest groups, the K-W test produced $p<1\times 10^{-10}$ (the last four bars in Figure 8). This underscores the significant differences between groups and sexes in each pair comparison.

Our research confirmed the individual differences in response to the warm stimulus. The warm-induced response in the WT group typically led to increased brain perfusion or recovery concerning the basal TOI in most pups, regardless of gender. In contrast, the L-NAME offspring showed no response (Figure 7 and Figure 8). The K-W test applied to normalized data for separate individuals depicted in Figure 7 yielded $p<1\times 10^{-10}$, confirming significant differences between the groups and underlining the importance of our research. These results further confirm alterations in the capacity of the brain microcirculation to detect and/or react to physical thermal stimuli. The physiological or pathophysiological implications of this differential response in offspring from PE require further investigation. However, we know that children born with acute hypoxia, including some infants born to women with PE, are candidates to receive therapeutic hypothermia. Based on our results in this scenario, we could hypothesize that this gold-standard treatment may not have similar outcomes in male and female offspring. However, whether or not this hypothesis is confirmed deserves further investigation.

To further validate our previous findings, we applied the pairwise multi-comparison test to the individual warm-to-cold differences, which we report in Table 2.

Table 2 shows the top nine individuals with remarkable variations in their warm-induced responses. These individuals were selected based on their unique responses compared to at least 27 other subjects we analyzed.

Notably, most mice are from the WT group, where six of them are male. These findings highlight the significant individual differences in response to the warm stimulus, a crucial aspect that deserves our interest. We rigorously tested these individual differences to determine whether they were due to specific individuals or the entire interest group. Table 2 reveals that several individuals have significant inter- and intra-group differences, indicating that these differences are not specific to individuals but to the entire interest groups. Indeed, excluding the three top individuals with significant differences with almost all the rest of the analyzed mice (№9, №16, and №5 in Table 1) did not affect statistical differences. We obtain the same results by analyzing the PLAs depicted in Figure 8.

Our comprehensive research has then provided significant insights into the warm-induced response in mice. It highlights individual variability and significant differences between groups and sexes. These findings underscore the importance and impact of our work in understanding the brain vascular perfusion response to diverse empiric interventions, such as thermal stimuli, in experimental models for PELS.

## 4. Discussion

Our general methodology, which relies on the PELT method for its high efficiency [19,23], is a reliable and accurate approach for robust brain perfusion signal analysis under various experimental settings. Our main contribution is that we have complemented it with careful data segmentation in optimal TOIs and a novel and suitable data normalization. We specifically looked to improve transition times detection, which motivated us to systematize, generalize, and significantly improve our previous methodology [18]. In addition, the research questions, whose answers are relevant to illuminate the underlying mechanisms of altered brain perfusion signals in L-NAME offspring compared with the uncomplicated WT offspring and the responsiveness of brain perfusion in L-NAME offspring to thermal stimuli, also motivated the enhancement of our methodology.

Our methodology is a step towards a systematic, accurate, and general approach to robust analysis of brain perfusion signals. We could further enhance it by considering other methodologies like those in the reference [24] to handle the transition times. In addition, we could improve our general methodology by incorporating algorithms for allowing multiscale changepoints, which may help us more accurately handle large jumps over short intervals like the transitions between empiric states [25]. Thus, we may extend the general methodology for accurately processing perfusion signals in broad experimental models to elucidate fundamental biological questions, as we did in the present study.

We have extended our previous evidence of reduced brain angiogenesis in the offspring of PELS [1] by demonstrating impaired microvascular reactivity in the brain. Specifically, offspring from L-NAME-treated dams lacked normal cold-induced vasoconstriction and warm-induced vasorecovery with respect to the basal state, indicating vascular dysfunction. We also found sex-specific differences: male WT offspring had a more significant drop in perfusion than females. In contrast, males in the L-NAME group exhibited higher perfusion after cold exposure than females. Individual variability did not affect the observed differences between WT and L-NAME groups or between sexes. Overall, our findings show that PE offspring underwent dysfunctional brain microvascular perfusion with inadequate responses to stimuli, which may lead to neuronal damage, neuroinflammation, and exacerbated hypoxia-induced brain injury.

While our findings suggest impaired vascular responses in offspring from preeclamptic dams, the absence of direct molecular assessment of endothelial dysfunction limits definitive conclusions. Nonetheless, previous studies have reported reduced brain angiogenesis and systemic endothelial alterations in similar models, supporting our interpretation [1,7,14,26,27,28].

Our findings are consistent with previous research showing that male infants born to preeclamptic pregnancies have reduced skin microvascular blood perfusion compared to those from normotensive pregnancies. In contrast, female infants do not show these differences [29]. Additional studies suggest endothelial-dependent vasodilation [30] and maximal capillary density is lower in term infants born to preeclamptic mothers [31]. These findings indicate that infants born to preeclamptic mothers, particularly males and those born at term, may have more pronounced systemic endothelial dysfunction than their counterparts from normotensive pregnancies.

Our study expands on these findings by examining brain microcirculation specifically. We noted a lack of variability in brain perfusion and vascular response to thermal stimuli in the offspring of L-NAME dams, consistent with our earlier reports [17,18]. Although we did not analyze the underlying mechanisms in this study, the alterations may also be associated with endothelial dysfunction, in this case, in the brain. Accordingly, previous studies in different PELS models have shown that offspring exhibit altered brain vasculature, such as smaller and less stiff middle cerebral arteries [20], potentially affecting blood flow autoregulation—the brain’s ability to maintain constant perfusion despite changes in blood pressure.

Sex differences were also evident in our study. Males exhibited more pronounced changes in brain perfusion responses than females, suggesting a sex-specific vulnerability to vascular dysfunction. While females have shown more significant cognitive impairments [32], male infants are often more susceptible to brain complications due to adverse perinatal outcomes, such as severe asphyxia or cerebral palsy [33,34]. Our results align with animal studies demonstrating greater susceptibility to cerebral complications in male offspring from PELS models [35,36]. For example, male offspring from reduced uterine perfusion pressure (RUPP) model dams showed significant brain edema compared to female siblings [17]. In the L-NAME model, we found an atypical response in male offspring: cold stimulus led to vasodilation instead of the expected vasoconstriction observed in males from the WT group or the lack of response seen in female L-NAME pups. This unexpected result suggests that standard treatments, like hypothermic therapy for hypoxic babies, may not be equally effective for both sexes.

In addition, increasing evidence indicates that offspring from PE have long-lasting consequences in brain function. That indicates a higher risk of developing cognitive impairments, including cerebral palsy, impaired neurodevelopment, or a high risk of behavioral alterations during adolescence [7,37]. However, the underlying mechanisms of this epidemiological, clinical, and preclinical evidence are unclear.

Maternal stress hormones, particularly elevated cortisol, have been implicated in fetal neurodevelopmental programming and may confound cerebrovascular outcomes in offspring [38,39]. While studies on maternal stress and cardiovascular programming exist, evidence directly linking maternal stress to cerebrovascular alterations in offspring remains scarce and warrants further investigation [40].

## 5. Conclusions

Our general methodology is a significant step towards a systematic, accurate, and reliable approach to robust brain perfusion signal analysis under various conditions and experimental settings. The results of the methodology reveal impaired brain perfusion in the offspring of preeclamptic pregnancies, underscoring the urgent need to better understand the underlying mechanisms. Altered brain vascular function may affect brain development in these children [15], suggesting that future research should focus on the sex-specific impacts of PE and endothelial dysfunction. This emphasis on sex-specific research could lead to more targeted interventions.

## Figures and Tables

**Figure 1 bioengineering-12-00675-f001:**
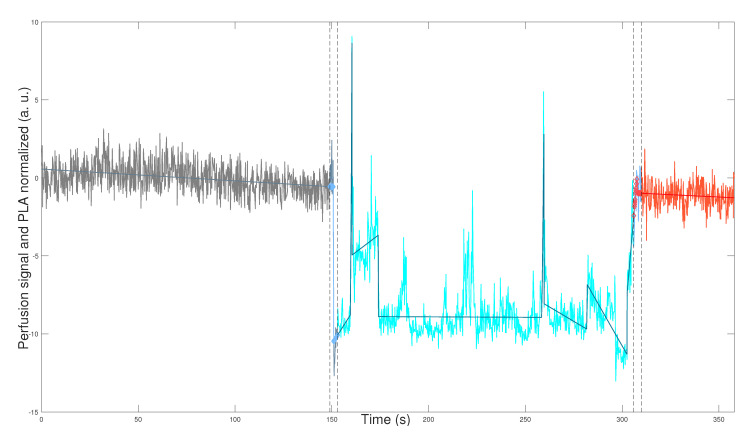
Normalized perfusion signals of a specific experimental subject. The normalized perfusion signals together with their PLAs, $z_{t}^{i}$ and ${\hat{z}}_{t}^{i}$, $t=1,\ldots ,n_{i}$, given by Equation (Equation 7). The light gray subplot corresponds to the basal state (optimal TOI j=1), the light blue to the cold state (optimal TOI j=2), and the light red to the warm state (optimal TOI j=3). Subject i=9 of the control group (WT male, internal code WT14D4).

**Figure 2 bioengineering-12-00675-f002:**
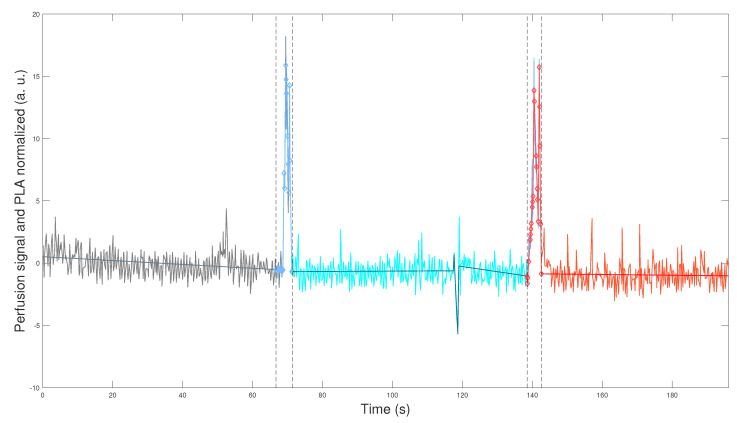
Normalized perfusion signals of a specific experimental subject. The normalized perfusion signals together with their PLAs, $z_{t}^{i}$ and ${\hat{z}}_{t}^{i}$, $t=1,\ldots ,n_{i}$, given by Equation (Equation 7). The same colors for the legends as in Figure 1 apply. Subject i=17 of the experimental group (L-NAME female, internal code L7D3).

**Figure 3 bioengineering-12-00675-f003:**
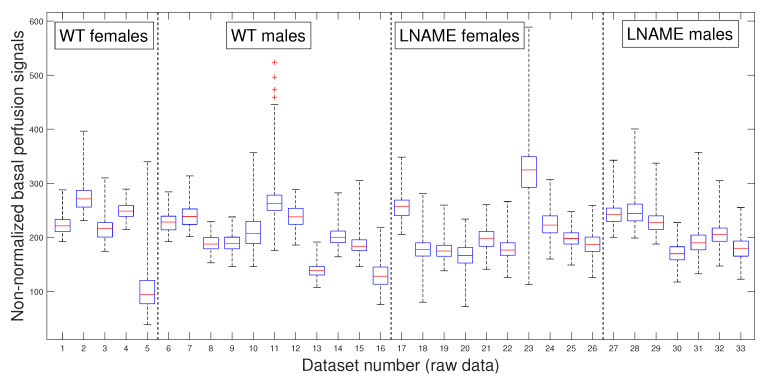
Box and whisker plots for WT and L-NAME perfusion signals. Non-normalized perfusion signal segment ${\left(y_{t}^{i,1}\right)}_{t=1,\ldots ,t^{i,1}}$ as defined by Equation (Equation 1) for optimal TOI j=1 or basal state. See Section B.5 for an explanation of box and whisker plots.

**Figure 4 bioengineering-12-00675-f004:**
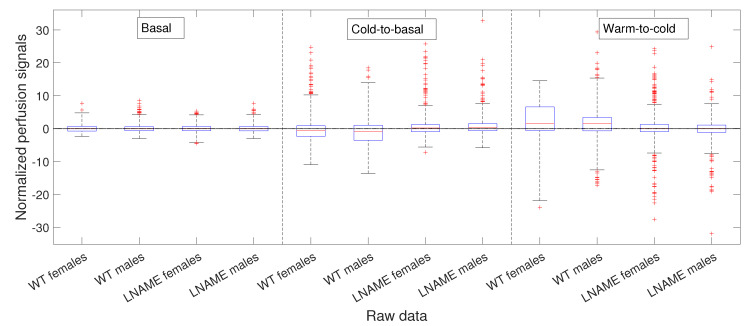
Box and whisker plots for WT and L-NAME perfusion signals. Box and whisker plots of $z_{t}^{i,1}$ in the basal state or TOI j=1 computed according to Equation (Equation 3), of $z_{t}^{i,2}-z_{t}^{i,1}$ in Equation (Equation 4) for cold-to-basal differences, and of $z_{t}^{i,3}-z_{t}^{i,2}$ in Equation (Equation 4) for warm-to-cold differences. See Section B.5 for an explanation of box and whisker plots.

**Figure 5 bioengineering-12-00675-f005:**
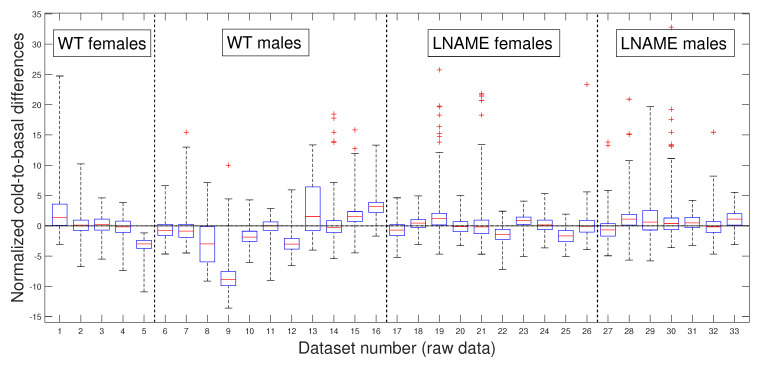
Box and whisker plots for WT and L-NAME cold-to-basal differences. Normalized perfusion signal differences $z_{t}^{i,2}-z_{t}^{i,1}$ for all i=1,…,N, defined in Equation (Equation 4). See Section B.5 for an explanation of box and whisker plots.

**Figure 6 bioengineering-12-00675-f006:**
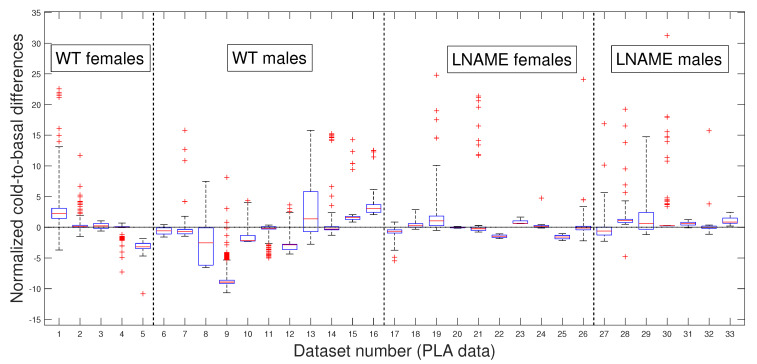
Box and whisker plots for WT and L-NAME cold-to-basal differences. Normalized perfusion signal’s PLA differences ${\hat{z}}_{t}^{i,2}-{\hat{z}}_{t}^{i,1}$ for all i=1,…,N, defined in Equation (Equation 4). See Section B.5 for an explanation of box and whisker plots.

**Figure 7 bioengineering-12-00675-f007:**
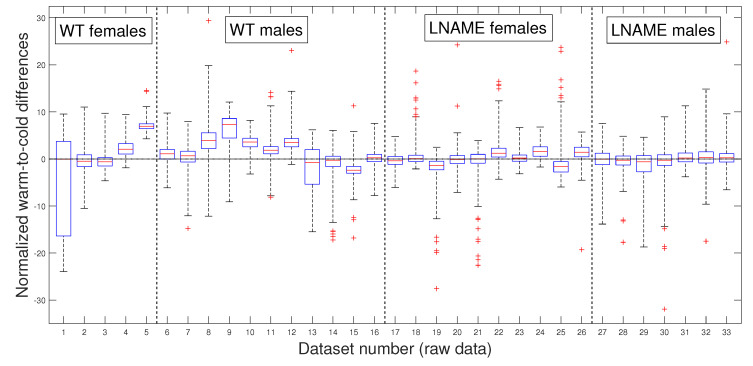
Box and whisker plots WT and L-NAME warm-to-cold differences. Perfusion signal differences $z_{t}^{i,3}-z_{t}^{i,2}$ for i=1,…,N, normalized according to Equation (Equation 3). See Section B.5 for an explanation of box and whisker plots.

**Figure 8 bioengineering-12-00675-f008:**
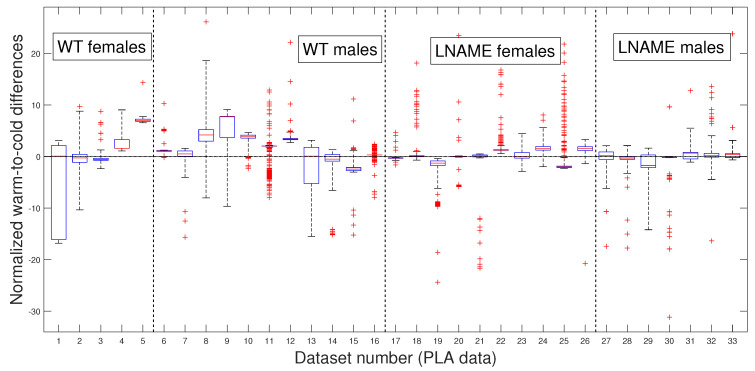
Box and whisker plots WT and L-NAME warm-to-cold differences. PLA differences ${\hat{z}}_{t}^{i,3}-{\hat{z}}_{t}^{i,2}$ for i=1…,N, normalized according to Equation (Equation 3). See Section B.5 for an explanation of box and whisker plots.

**Table 1 bioengineering-12-00675-t001:** Individual differences in response to the cold stimulus.

№	Code	Group of Interest	Overall Differences	Differences with WT	Differences with L-NAME
9	WT14D4	WT male	32: 15 WT and 17 LNAME	All others	All
16	WT16D5	WT male	32: 15 WT and 17 LNAME	All others	All
5	WT15D3 ^1^	WT female	31: 14 WT and 17 LNAME	All others except №12	All
12	WT15D4 ^1^	WT male	31: 14 WT and 17 LNAME	All others except №5	All
8	WT14D1 ^2^	WT male	29: 14 WT and 15 LNAME	All others except №10	All except №22 and №25
10	WT14D5 ^2^	WT male	29: 14 WT and 15 LNAME	All others except №8	All except №22 and №25
22	L12D4 ^3^	LNAME female	29: 14 WT and 15 LNAME	All except №8 and №10	All others except №25
25	L13D4 ^3^	LNAME female	29: 14 WT and 15 LNAME	All except №8 and №10	All others except №22
17	L7D3	LNAME female	28: 12 WT and 15 LNAME	All except №4, №6 and №7	All others except №27
15	WT16D4	WT male	2: 13 WT and 14 LNAME	All others except №1 and №13	All except №19, №28 and №33

^1^ Individuals responded similarly to the cool stimulus. ^2^ Individuals responded similarly to the cool stimulus. ^3^ Individuals responded similarly to the cool stimulus.

**Table 2 bioengineering-12-00675-t002:** Individual differences in response to the warm stimulus.

№	Code	Group and Sex	Overall Differences	Difference with WT	Difference with L-NAME
5	WT15D3	WT female	31: 14 WT and 17 L-NAME	All others except №9	All
15	WT16D4	WT male	31: 14 WT and 17 L-NAME	All others except №1	All
9	WT14D4	WT male	30: 13 WT and 17 L-NAME	All others except №5 and №12	All
19	L11D2	L-NAME female	30: 15 WT and15 L-NAME	All except №1	All others except №25
8	WT14D1 ^1^	WT male	29: 12 WT and 17 L-NAME	All except №4, №10 and №12	All
10	WT14D5 ^1^	WT male	29: 12 WT and 17 L-NAME	All except №4, №8, and №12	All
12	WT15D4	WT male	29: 12 WT and 17 L-NAME	All except №8-10	All
25	L13D4	L-NAME female	28: 14 WT and 14 L-NAME	All except №1 and №3	All others except №19 and №29
11	WT15D1	WT male	27: 13 WT and 14 L-NAME	All others except №1 and №4	All except №22, №24, and №26

^1^ Individuals responded similarly to the warm stimulus.

## Data Availability

The datasets studied in the work are included in the Appendix A. The corresponding authors can be contacted for further inquiries.

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
