# Peer review of "A Changepoint Detection-Based General Methodology for Robust Signal Processing: An Application to Understand Preeclampsia’s Mechanisms"

_bioengineering, 2025, doi:10.3390/bioengineering12060675_

Round 1

Reviewer 1 Report

Comments and Suggestions for Authors

INTRODUCTION & RELATED LITERATURE:

  1. Introduction claims only eight PELT citations but omits search date, database filters, and keywords, hurting transparency.
  2. Cited epidemiology of PE in offspring is outdated and misses 2024 meta-analyses.
  3. Literature section fails to critique prior PLA-based segmentation accuracy on synthetic datasets.

METHODS:

  1. Methods do not provide animal randomization or blinding details.
  2. Algorithm 1 sets constant k but never justifies its numeric value relative to sampling frequency.
  3. Kruskal–Wallis use is appropriate yet assumptions of equal shape distributions are not checked.
  4. Statistical plan does not state alpha correction for 50+ pairwise tests.

RESULTS & DISCUSSIONS:

  1. Results discuss vasodilation yet never show time-resolved traces proving recovery kinetics.
  2. Discussion attributes dysfunction to endothelial issues without molecular validation.
  3. Claims of microglial activation are speculative and uncited.
  4. Discussion does not consider confounding maternal stress hormones on offspring vasculature.
  5. No comparison with alternative segmentation frameworks (e.g., Bayesian online changepoint) is offered.
  6. Results describe Figure 1 but miss scale bar units on perfusion axis 

CONCLUSIONS:

  1. Conclusion repeats “microvascular” five times, suggesting need for concision.
  2. Conclusion restates results but provides no actionable future experiment plan.

Author Response

Please, see the pdf file.

Reviewer 2 Report

Comments and Suggestions for Authors

The paper presents a signal processing methodology applied to the study of preeclampsia mechanisms through analysis of brain perfusion signals. The approach is relevant and addresses a significant clinical challenge. The manuscript is generally well written, with a thorough methodology, clearly presented results, and conclusions that are appropriately supported by the data.

Minor Edit-Revisions Required Before Publication:

1. In the abstract, the term preeclampsia (PE) is defined, but the abbreviation PE is not subsequently used (e.g., in the phrase "preeclampsia-like syndrome"). Either make consistent use of the abbreviation throughout or remove it from the abstract entirely. Additionally, the abbreviation PE is redefined in the main text (line 214). Please ensure that all abbreviations are defined only upon their first use and are used consistently thereafter.

2. Please remove novel type reading (careully) ... To carefully calculate optimal segmentation

3. While not mandatory, it would enhance readability and comprehension if the material currently placed in Appendices A and B were incorporated into the main text. This would reduce the need to switch between sections when following the methodological details.

Author Response

Please, see the pdf file.

Reviewer 3 Report

Comments and Suggestions for Authors

The manuscript deals with a very important topic - determining how to control pre-eclampsia to produce healthy offspring. The experiments were conducted on rodents. The authors have done quite a lot of work. The manuscript certainly deserves special attention. However, I believe that it should not be published in this form. 
The manuscript is very difficult to read. I attribute this to several reasons: 
Even to define the key concepts on which the whole manuscript is based, I have to refer to the reference literature. For example, how TOI is defined, I had to look up in the authors' previous papers (line 10).
The manuscript is very strangely structured. The appendices contain sections that I consider particularly important. For example, Appendix A includes an experimental model. And in the place where the experimental model should be, a reference to it is left. From the reader's point of view, this is very inconvenient. 
Dear authors, please explain in the text of the article what is perfusion signal, how it is located, in what units it is measured (Fig. 1 and 2)? 
Why are there three identical rows in Table 1?
Why are strong conclusions drawn on such a small sample? 
Figure 2 is only half filled. Please use the entire area of the figure.
You state that your methodology represents a significant advance in the analysis of perfusion signals, yet no numerical results are presented in the manuscript. How can this be the case?

Author Response

Please, see the pdf file.

Round 2

Reviewer 1 Report

Comments and Suggestions for Authors

1. All queries are answered correctly and promptly.

Reviewer 3 Report

Comments and Suggestions for Authors

I thank the authors of the manuscript for their comprehensive answers. A very interesting article indeed. I have no further questions. I have no objection to the article being published in this esteemed journal.